# Crosstalk between BMP and Notch Induces Sox2 in Cerebral Endothelial Cells

**DOI:** 10.3390/cells8060549

**Published:** 2019-06-06

**Authors:** Xiuju Wu, Jiayi Yao, Lumin Wang, Daoqin Zhang, Li Zhang, Eric X. Reynolds, Tongtong Yu, Kristina I. Boström, Yucheng Yao

**Affiliations:** 1Division of Cardiology, David Geffen School of Medicine at UCLA, Los Angeles, CA 90095-1679, USA; XiujuWu@mednet.ucla.edu (X.W.); jyao@mednet.ucla.edu (J.Y.); wanglumin1@126.com (L.W.); daoqinzhang@mednet.ucla.edu (D.Z.); LiZ@mednet.ucla.edu (L.Z.); ericx02@icloud.com (E.X.R.); tongtongyu@mednet.ucla.edu (T.Y.); 2Department of cell Biology and Genetics, School of Basic Medical Sciences, Xi’an Jiaotong University Health Science Center, Xi’an, 710061, China; 3Department of Cardiology, Shengjing Hospital of China Medical University, Shenyang 110004, China; 4The Molecular Biology Institute at UCLA, Los Angeles, CA 90095-1570, USA

**Keywords:** Sox2, bone morphogenetic protein, Notch, endothelial cells

## Abstract

Bone morphogenetic protein (BMP) and Notch signaling are critical for endothelial cell (EC) differentiation in vascular development. Recent studies have shown that excess BMP activity induces Notch signaling in cerebral ECs resulting in arteriovenous malformation (AVMs). However, it is unclear how the crosstalk between BMP and Notch signaling affects cerebral EC differentiation at the gene regulatory level. Here, we report that BMP6 activates the activin receptor-like kinase (ALK) 3, a BMP type 1 receptor, to induce Notch1 receptor and Jagged1 and Jagged2 ligands. We show that increased expression of the Notch components alters the transcriptional regulatory complex in the SRY-Box 2 (*Sox2*) promoter region so as to induce its expression in cerebral ECs. Together, our results identify Sox2 as a direct target of BMP and Notch signaling and provide information on how altered BMP and Notch signaling affects the endothelial transcriptional landscape.

## 1. Introduction

The bone morphogenetic protein (BMP) and Notch signaling pathways are essential for vascular development and homeostasis, and disruption of these pathways is known to cause vascular disease. Mutations of the activin receptor-like kinase (ALK) 1, a BMP type I receptor, are linked to hereditary hemorrhagic telangiectasia 2 (HHT2), which is characterized by the presence of arteriovenous malformation (AVMs) in multiple organs including the brain [1,2]. Excess BMP4 and BMP6 activities are known to contribute to cerebral vascular malformation [3,4], and loss of the BMP inhibitor matrix Gla protein (MGP) causes AVMs in brain, lungs, and kidneys, similar to HHT2 [4,5]. Notch components are considered to be critical mediators of endothelial cell fate decisions and vascular lumen formation [6,7], and both loss-of-function and gain-of-function Notch mutations result in arteriovenous shunting [8,9]. Constitutively active Notch4, in particular, causes brain AVMs [10,11]. Recent studies from our lab showed that lack of BMP inhibition induces the expression of Notch components in endothelial cells (ECs), which results in cerebral AVMs [4,12]. However, it is poorly understood how the interactions between BMP and Notch signaling alter transcriptional regulation in cerebral ECs, contributing to vascular disease in the brain. 

Sox2 belongs to the Sox family of transcription factors, characterized by a DNA-binding high mobility group (HMG) box domain [13,14]. Sox2 is a well-known cell fate regulator and functions as an important transcription factor for tissue development [15] and cell reprogramming [14]. Sox2 is also involved in cell–cell transitions, such as epithelial–mesenchymal transitions [16,17] and endothelial–mesenchymal transitions [18]. Here, we report that interactions between BMP and Notch signaling induce Sox2 expression in cerebral ECs. We show that BMP6 activates the BMP type 1 receptor ALK3, increasing the expression of specific Notch ligands and of a Notch receptor, that in turn activate the expression of Sox2 in cerebral ECs. 

## 2. Materials and Methods

### 2.1. Animals 

*Mgp^+/−^* (B6.129S7-Mgptm1Kry/KbosJ), *Jagged1^+/−^ (B6.129S1-Jag1tm1Grid/J)*, *Jagged2^+/−^* (B6.129S1-Jag2tm1Grid/J), *Cdh5^Cre^* (B6.Cg-Tg(Cdh5-cre)7Mlia/J), and *Notch1^Flox/Flox^* (Notch1tm2Rko/GridJ) mice on a C57BL/6J background were obtained from the Jackson Laboratory. The genotypes were confirmed by PCR [19], and the experiments were performed with generations F4–F6. Littermates were used as wild-type controls. All mice were fed a standard chow diet (Diet 8604, HarlanTeklad Laboratory). The studies were reviewed and approved by the Institutional Review Board and conducted in accordance with the animal care guidelines set by the University of California, Los Angeles. The investigation conformed to the National Research Council, *Guide for the Care and Use of Laboratory Animals, Eighth Edition* (Washington, DC: The National Academies Press, 2011).

### 2.2. Tissue Culture

Human brain microvascular cells (HBMECs) were obtained from ScienCell Research Laboratories and cultured as per the manufacturer’s protocol. For treatment, Jagged1 and 2 (R&D Systems) were added as indicated in the Results section. Transient transfections of HBMECs with siRNA (Silencer® predesigned siRNA, Applied Biosystem, Foster City, CA, USA) were optimized and performed as previously described [5]. When compared to unrelated control siRNA and scrambled siRNA, the selected siRNAs resulted in a 90–95% decrease in mRNA and protein levels, as determined by real-time PCR and immunoblotting, respectively. Silencer® predesigned siRNAs were obtained for ALK2, 3, 4, 6, and 7 and Notch1, 2, 3, and 4.

### 2.3. Isolation of Cerebral ECs

Cerebral EC were isolated as previous described [20]. Briefly, mouse cerebra were collected after systemic perfusion with phosphate-buffered saline (PBS). The cerebra were transferred to a Petri dish, cut into small pieces, and treated with 0.5% collagenase. The mixture was incubated at 37 °C with slow rotation for 30 minutes. After the incubation, a Pasteur pipette was applied with gentle pressure to disaggregate the tissues. The cell suspension was filtered through a 100 µm mesh into a new tube, and centrifuged at 300× *g* for 5 minutes at 4 °C. The cell pellet was re-suspended in PBS with 2% BSA for incubation with primary CD31 and CD45 antibodies to label the cells for flow-cytometric sorting. CD31^+^CD45^−^ cells were collected for EC experiments, and the remaining cells were used as controls.

### 2.4. RNA Analysis

Real-time PCR analysis was performed as previously described [21]. Glyceraldehyde 3-phosphate dehydrogenase (*GAPDH*) was used as a control gene [21]. Primers and probes for mouse or human Jagged1, *Jagged2*, *Notch* 1–4, *Dll 1*, *3*, and *4*, and *Sox2* were obtained from Applied Biosystems as part of TaqMan Gene Expression Assays.

### 2.5. Immunoblotting 

Immunoblotting and immunoprecipitation were performed as previously described [22]. Equal amounts of tissue lysates were used for immunoblotting. Blots were incubated with specific antibodies to Sox2 (Abcam, Cambridge, MA, USA, ab97959). Beta-Actin (1:5000 dilution; Sigma-Aldrich, St. Louis, MO, USA) was used as a loading control. 

### 2.6. Chromatin Immunoprecipitation (ChIP) Assay

ChIP assays were performed as previously described [23]. Briefly, glycine was added to the cells to a concentration of 0.125 M to quench the crosslinking. The cells were then rinsed with ice-cold PBS, re-suspended, lysed in lysis buffer, and sonicated to shear the crosslinked DNA to fragments ranging from 200 to 500 bp, as previously described [24]. The lysates ere incubated with specific antibodies or normal IgG at 4 °C overnight. After adding 40 μL of protein G magnetic beads, the lysates were further incubated for 2–3 hours. The beads were washed repeatedly, and the DNA was eluted from the beads by incubating in 10 mM Tris-Cl, pH 8.5, for 15 minutes at 65 °C. Both the immunoprecipitated and the input DNA samples were incubated overnight at 65 °C for reversal of the crosslinking. The DNA samples were then purified by sequential phenol/chloroform/isoamyl alcohol (Sigma) extraction. The final DNA products were ethanol-precipitated, and the pellets were air-dried and dissolved in 10 mM Tris-HCl. The primers were used for the ChIP assay as previously described [25]. Anti-RBPJκ (Abcam, ab25949), anti-NICD1 (Abcam, ab83232), and anti-MAM (Abcam, ab17019) antibodies were used for the ChIP assays. 

### 2.7. ChIP-seq

ChIP-seq of DNA from *Mgp^−/−^ and Mgp^+/+^* cerebral ECs were performed using specific anti-H3K4me3 (Abcam, rabbit ab8580) and anti-H3K27me3 (Abcam, mouse ab6002) antibodies in order to enrich the genomic DNA. ChIP DNA was sequenced by the Technology Center for Genomics and Bioinformatics at UCLA. Reads from each sample were mapped to the human genome by using Bowtie2. The HOMER tool was used to detect a significant enrichment of peaks with 5% false discovery rate and more than four-fold over-input. Motif occurrences in peaks were identified by The HOMER Motif Discovery function. Peak annotation was performed to associate peaks with nearby genes to calculate tag densities. 

### 2.8. Statistical Analysis

Data were analyzed for statistical significance by ANOVA with post-hoc Tukey’s analysis. The analyses were performed using GraphPad Instat®, version 3.0 (GraphPad Software, San Diego, CA, USA). Data represent mean ± SD; *p*-values less than 0.05 were considered significant, and experiments were repeated a minimum of three times.

## 3. Results

### 3.1. Excess BMP Activity Induces Notch in Cerebral ECs

MGP is an inhibitor of BMP 2, 4, 6, and 7, and the MGP-deficient (*Mgp^−/−^*) mouse is a well-documented model of cerebral AVMs. To determine which Notch ligands and receptors are induced by excessive BMP signaling, we isolated *Mgp^−/−^* cerebral ECs (Figure 1a) and examined the components of the Notch signaling pathway, including Notch receptors 1, 2, 3, and 4, Notch ligands Jagged1 and 2 and Delta-like (Dll) 1, 3, and 4. The results showed that the expression of Notch1 and Jagged1 and 2 was significantly increased in *Mgp^−/−^* cerebral ECs, as determined by real-time PCR (Figure 1b), suggesting that these Notch components were BMP downstream targets in cerebral ECs. 

### 3.2. BMP6 Activates ALK3 to Induce Notch in Brain ECs

To identify which BMP ligand and receptor were responsible for the induction of the Notch components, we treated HBMECs with different doses of BMP ligands (0–300 ng/mL), including BMP2, 4, 6, 7, 9, and 10, according to previous studies [5,19,26,27,28,29,30]. After 48 h of treatment, we examined the expression of Notch1 and Jagged1 and 2 and found that BMP6 strongly induced these proteins (Figure 2a). In addition, BMP9 mildly induced the same Notch components (Figure 2a). To further differentiate the responses, we treated HBMECs with BMP6 (100 ng/mL) and BMP9 (10 ng/mL). The results showed that BMP9 induced all the ligands and receptors of the Notch pathways, similar to what observed in our previous study [4], whereas BMP6 only induced the expression of Notch1 and Jagged1 and 2 (Figure 2a,b). Indeed, the BMP6 induction patterns of Notch1 and Jagged1 and 2 in HBMECs were similar to the induction patterns in the cerebral endothelium of *Mgp^−/−^* mice. 

Furthermore, we treated HBMECs with BMP6 and examined the phosphorylation (p) of SMAD 1/5/8 after 30 and 60 min of treatment. The results showed that pSMAD1/5/8 was significantly increased at 30 min and even more at 60 min (Figure 2c), suggesting that the early response to BMP6 in HBMECs was mediated by pSMAD1/5/8. We then examined the Notch1 intracellular domain (NICD1) and found that NICD1 was significantly increased after 5 and 10 h of BMP6 treatment in HBMECs (Figure 2d). The results suggested that BMP6 induced the Notch components through pSMAD1/5/8.

To assess BMP expression in *Mgp^-/-^* brains, we examined the levels of BMPs using real-time PCR and compared them to those of wild-type brain. The results showed no significant difference between wild-type and *Mgp^−/−^* brains (Figure 2e). The results were consistent with those of our previous studies [5,27,29], which showed that the lack of MGP increased BMP activity but not necessarily its expression. Together, the results suggested that excess activity of BMP6 in cerebral ECs caused the induction of Notch1 and Jagged1 and 2. 

To determine which receptor was activated by BMP6 to induce Notch, we individually depleted ALK-2, 3, 4, 6, and 7 in BMP6-treated HBMECs using specific siRNAs. The selected siRNAs decreased the targeted mRNA by 90%–95% compared to scrambled siRNA, as determined by real-time PCR (Figure 3a). The results showed that the depletion of ALK3 abolished Notch induction in BMP6-treated HBMECs (Figure 3b–d), suggesting that BMP6 activated ALK3 to induce Notch1 and Jagged 1 and 2 through ALK3 signaling. 

### 3.3. Elevated Notch Signaling Induces the Expression of Sox2

Our previous studies showed that lack of BMP inhibition increased Sox2 in the aortic endothelium and altered endothelial cell fate to contribute to vascular calcification [18]. To determine if Notch signaling affected Sox2 expression in brain ECs, we treated HBMECs with Jagged1, Jagged2, or their combination. We found that Jagged1 and 2 strongly induced Sox2 expression, and the induction was even higher when they were added in combination (Figure 4a,b). NICD1 was used as a control to show the activation of the Notch1 pathway (Figure 4b). To identify which Notch receptor was activated by Jagged1 and 2 to induce Sox2 expression, we individually depleted Notch1, 2, 3, and 4 in HBMECs treated with Jagged1 and 2 (Figure 4c,d) and examined Sox2 expression. The results of both real-time PCR and immunoblotting showed that the depletion of Notch1 abolished the induction of Sox2 (Figure 4c,d), suggesting that Jagged1 and 2 activated Notch1 to induce Sox2 expression.

To confirm our results in vivo, we limited Notch1 expression in *Mgp^−/−^* ECs by breeding VE-cadherin (Cdh5) promoter-driven Cre transgenic mice (*Cdh5^cre^*) with *Notch1^Flox/−^* mice. We isolated cerebral ECs from *Cdh5^cre^Notch1^Flox/−^Mgp^−/−^* mice and assessed the expression of Notch1 and Sox2. The results showed that limiting Notch1 abolished the induction of Sox2 in *Mgp^−/−^* cerebral ECs (Figure 4e). We also examined Sox2 expression in cerebral ECs of *Jagged1^+/−^Mgp^−/−^* and *Jagged2^+/−^Mgp^−/−^* mice with reduced Jagged1 and 2 [4]. The results showed that limiting Jagged1 and 2 abolished the induction of Sox2 in *Mgp^−/−^* cerebral ECs (Figure 4f). Together, the results showed that excess BMP6 activated ALK3 to induce Notch1 and Jagged1 and 2, which in turn induced the expression of Sox2 in cerebral ECs.

### 3.4. Excess Notch Signaling Alters the Regulatory Complex to Activate Sox2 Expression 

To determine how excess Notch signaling induced Sox2, we examined the binding of the recombination signal binding protein for immunoglobulin kappa J (RBPJκ) in the *Sox2* promoter. RBPJκ was reported to be activated by Notch signaling and recruit other Notch-associated factors to regulate the transcription of Notch target genes [31]. There are five RBPJκ-binding sites located in a 5 kb region upstream of the *Sox2* promoter (Figure 5a) [25]. By utilizing specific antibodies to RBPJκ, we performed chromatin immunoprecipitation (ChIP) to enrich RBPJκ-bound genomic DNA from *Mgp^+/+^ and Mgp^−/−^* cerebral ECs and examined the actual RBPJκ binding around the five binding sites. The results showed that DNA binding by RBPJκ strongly increased at the binding sites 1, 2, and 5 in *Mgp^−/−^* ECs (Figure 5b), confirming that increased Notch signaling altered the transcriptional regulation of the *Sox2* gene.

To determine the status of chromatin around the *Sox2* promoter in *Mgp^−/−^* cerebral ECs, we performed a ChIP in parallel with massive sequencing (ChIP-seq). We examined the bivalent marks trimethylated histone H3 lysine 4 (H3K4me3) and trimethylated histone H3 lysine 27 (H3K27me3). H3K4me3 is associated with active transcription, and H3K27me3 is associated with closed chromatin [32]. The results showed a significant increase of H3K4me3 and a decrease of H3K27me3 around the *Sox2* promoter in *Mgp^−/−^* ECs (Figure 5c), strongly indicating *Sox2* transcriptional activation. 

Furthermore, we examined the abundance of NICD1 and Notch-associated protein mastermind (MAM) around RBPJκ binding site 4 in the *Sox2* promoter in ECs from *Mgp^−/−^*, *Jagged1^+/−^Mgp^−/−^*, and *Jagged2^+/−^Mgp^−/−^* mice. Real-time PCR showed significant decreases in NICD1 and MAM binding in ECs from the *Jagged1^+/−^Mgp^−/−^* and *Jagged2^+/−^Mgp^−/−^* mice (Figure 5d), further confirming that increased Notch signaling modified the complex of Notch-associated proteins in the *Sox2* promoter.

Thus, we have identified BMP6 as an inducer of Notch components that alters the complex of Notch-associated proteins in the *Sox2* promoter and activates *Sox2* expression in cerebral ECs (Figure 5e).

## 4. Discussion

BMP and Notch signaling interact in many developmental processes, such as heart development [33] and osteogenic differentiation [34]. The interplay between BMP and Notch signaling may alter the downstream effects of BMP [35,36,37] or change Notch activity on target genes [35,36,37], at least in part through the interaction between NICDs and SMADs [36,37]. Interactions between BMP and Notch signaling are also found in disease, such as cancer [38], pulmonary arterial hypertension [39], and valvular calcification [40]. Our previous studies showed that loss of the BMP inhibitor MGP enhanced BMP signaling and induced Notch signaling in cerebral ECs, resulting in cerebral AVMs [4]. However, the BMP ligands and receptors involved in this crosstalk were unclear. Here, we found that BMP6/ALK3 directly induced the Notch ligands Jagged1 and 2 and the Notch receptor Notch1 in brain ECs. The finding differs from the BMP9/ALK1 induction of all components previously described [4] and also shown in this study (Figure 2) and provides a new regulatory machinery in cerebral EC differentiation.

Interactions between Sox2 and BMPs have been found in several cases during cell fate determination, including the differentiation of epithelial stem cells in molars and incisors [41], corneal endothelial cells [42], and gastric cells [43]. Moreover, loss of BMP inhibition, such as in MGP deficiency, induces Sox2 and triggers ECs to undergo endothelial–mesenchymal transitions [18]. It is possible that Sox2 and BMPs modulate mutual transitions of endothelium and mesenchyme to achieve cell differentiation, as well as the coordination between tissue-specific elements and vasculature. 

Although the interaction between Sox2 and Notch signaling in ECs constitutes a new field of investigation, previous studies have indicated the possibility of this connection. In fact, both Notch and Sox2 are involved in epithelial–mesenchymal transitions [16,44] and in the specification of sensory cell progenitors in the ear [45]. Notch regulates Sox2 in neural stem cells [25] and cochlear development [46], and the levels of Sox2 affect Notch signaling in tumorigenesis [47]. Here, our results provide a new link between Notch and Sox2 in EC differentiation and vascular formation. We showed that Sox2 is a direct target of Notch signaling in brain ECs and that the DNA binding of RBPJκ in the *Sox2* promoter is elevated with excess BMP activity. Since pSMAD1/5/8 are also increased in *Mgp^−/−^*-null ECs, the possibility remains that RBPJκ interacts with the SMADs to regulate Sox2 expression. Together, our findings enhance our understanding of cerebral EC differentiation on a transcriptional level as the basis of cerebral vascular disease.

## Figures and Tables

**Figure 1 cells-08-00549-f001:**
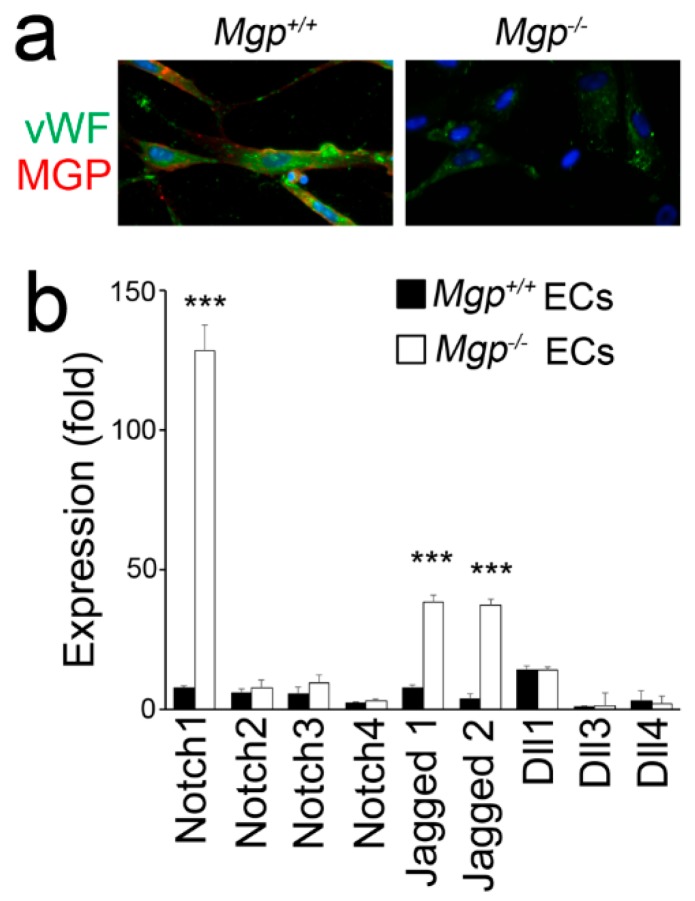
Increased Notch signaling in *Mgp^−/−^* cerebral endothelial cells (ECs). (**a**) Immunostaining of cerebral ECs isolated from wild-type (*Mgp^+/+^*) and *Mgp^−/−^* mice. The cells were incubated for 24 hours to let them attach to the slides before staining; vWF, von Willebrand factor. (**b**) Expression of Notch receptors 1, 2, 3, and 4, and Notch ligands Jagged1 and 2 and Delta-like (Dll)1, 3, and 4 in cerebral ECs isolated from wild-type (*Mgp^+/+^*) and *Mgp^−/−^* mice (n = 8), as determined by real-time PCR. *** *p* < 0.001.

**Figure 2 cells-08-00549-f002:**
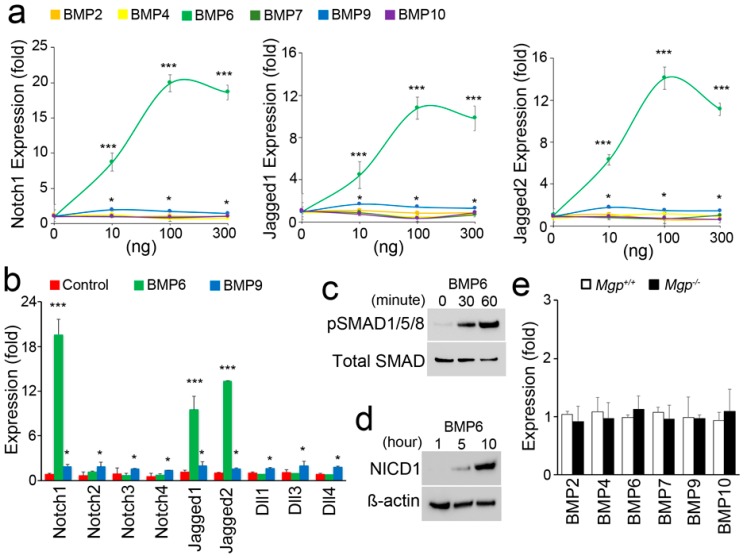
Bone morphogenetic protein 6 (BMP6) induces Notch1 and Jagged1 and 2 in cerebral ECs. (**a**) Expression of Notch1 and Jagged1 and 2 in human brain microvascular cells (HBMECs) treated with different doses of BMP2, 4, 6, 7, 9, or 10. (**b**) Expression of components of the Notch pathway in HBMECs treated with BMP6 (100 ng/mL) and BMP9 (10 ng/mL). (**c**) Immunoblotting of pSMAD1/5/8 in HBMECs at 0, 30, and 60 min of treatment with BMP6. (**d**) Immunoblotting of Notch1 intracellular domain (NICD1) in HBMECs at 1, 5, and 10 h of treatment with BMP6. (**e**) Expression of BMPs in cerebra (n = 5). * *p* < 0.05; *** *p* < 0.001.

**Figure 3 cells-08-00549-f003:**
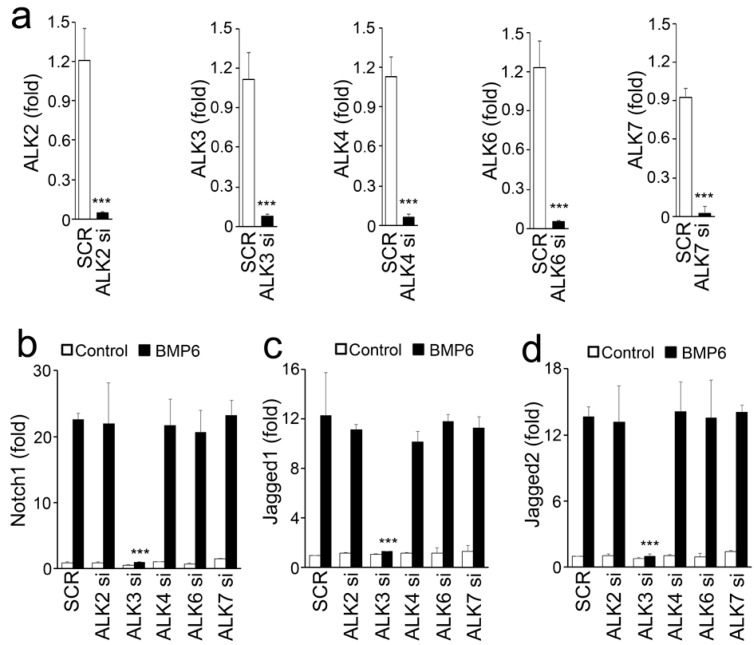
BMP6 induces Notch signaling through the activation of ALK3. (**a**) Expression of ALK2, 3, 4, 6, and 7 in HBMECs after transfection with individual siRNAs. (**b**–**d**) Expression of Notch1 (**b**), Jagged1 (**c**) and 2 (**d**) in BMP6-treated HBMECs after transfection with specific siRNAs (si) to ALK2, ALK3, ALK4, ALK6, or ALK7. *** *p* < 0.001.

**Figure 4 cells-08-00549-f004:**
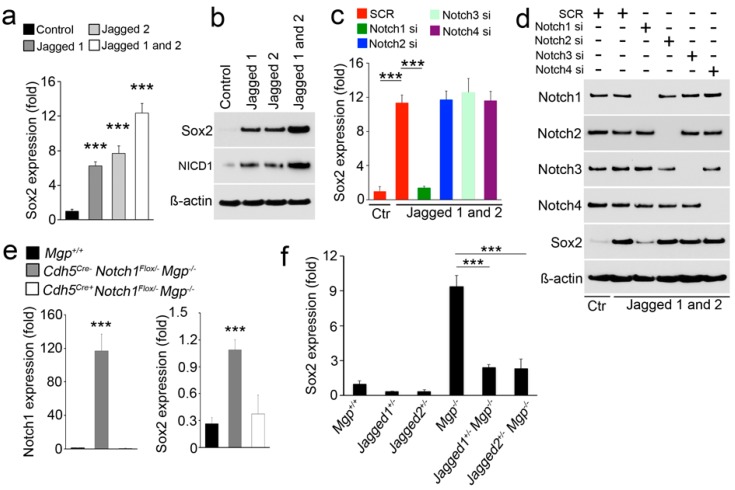
Excess Notch signaling induces Sox2 in cerebral ECs. (**a**,**b**) Sox2 expression in HBMECs treated with Jagged1, Jagged2, or their combination, as detected by real-time PCR (**a**) and immunoblotting (**b**). NICD1 was used as a control for the activation of Notch1 signaling. (**c**,**d**) Sox2 expression in HBMECs treated with Jagged1 and 2 after transfection with specific siRNAs (si) to Notch1, 2, 3, or 4 detected by real-time PCR (**c**) and immunoblotting (**d**). Ctr: control. (**e**) Expression of Notch1 and Sox2 in *Cdh5^Cre−^Notch^Flox/−^Mgp^−/−^* and *Cdh5^Cre+^Notch^Flox/−^Mgp^−/−^* cerebral ECs. Wild-type ECs (*Mgp^+/+^*) were used as a control (n = 6). (**f**) Expression of Sox2 in cerebral ECs isolated from wild-type, *Jagged1^+/−^*, *Jagged2^+/−^*, *Mgp^−/−^*, *Jagged1^+/−^Mgp^−/−^*, and *Jagged2^+/−^Mgp^−/−^* mice. *** *p* < 0.001.

**Figure 5 cells-08-00549-f005:**
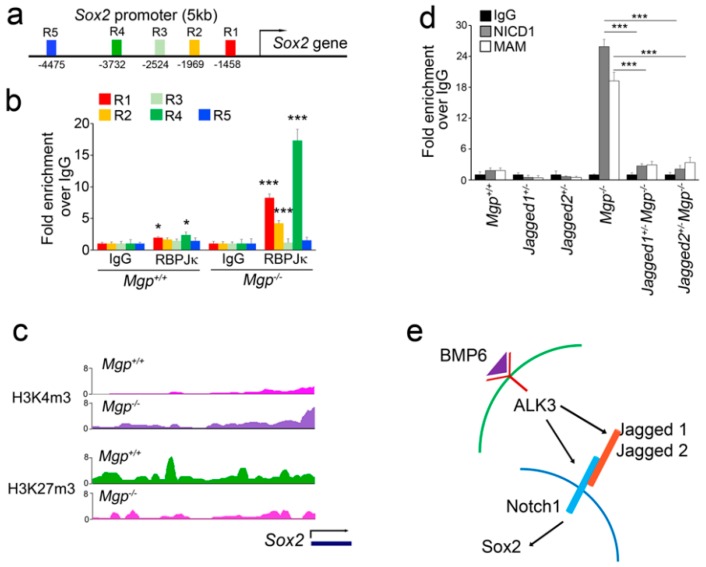
Notch signaling alters the regulatory complex in the *Sox2* promoter to activate its expression in Mgp^-/-^ cerebral ECs. (**a**) Recombination signal binding protein for immunoglobulin kappa J (RBPJκ) binding (R) sites in +5 kb upstream of the regulatory region of the *Sox2* gene. (**b**) Chromatin immunoprecipitation (ChIP) assays demonstrating increased RBPJκ binding in the *Sox2* regulatory region in *Mgp^−/−^* cerebral ECs. (**c**) ChIP-seq data demonstrating the occupation of the bivalent marks trimethylated histone H3 lysine 4 (H3K4me3) and trimethylated histone H3 lysine 27 (H3K27me3) around the *Sox2* gene locus in *Mgp^+/+^* and *Mgp^−/−^* cerebral ECs. (**d**) ChIP assays demonstrating the increased binding of NICD1 and mastermind (MAM) around RBPJκ binding site 4 in the *Sox2* regulatory region of *Mgp^−/−^* cerebral ECs and the abolished binding in *Jagged1^+/−^Mgp^−/−^* and *Jagged2^+/−^Mgp^−/−^* cerebral ECs. * *p* < 0.05; *** *p* < 0.001. (**e**) Schematic diagram.

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
