# Peer review of "Crosstalk between BMP and Notch Induces Sox2 in Cerebral Endothelial Cells"

_cells, 2019, doi:10.3390/cells8060549_

Round 1

Reviewer 1 Report

In this manuscript, Wu et al. described 1) Increased Notch signaling in cerebral ECs from Mgpknockout mice; 2) BMP6 activates ALK3 to induce Notch signaling in human brain macrovascular cells; 3) Elevated Notch signalling induces the expression of Sox2; 4) Excess Notch signaling alters the regulatory complex to activate Sox2 expression. It is the follow up work of previous publications from the same group that reduced Jagged 1 and 2 levels prevents cerebral arteriovenous malformations in Mgp -/- mice (ref 4) and Sox2 expression is increased in mice with EC specific deletion of Mgp. The current manuscript is aiming to establish that the crosstalk between BMP and Notch signalling affects EC differentiation at the gene regulatory level. Overall the manuscript describes an interesting result important in endothelial biology, but it lacks some important controls to support the major conclusion. 

Major points:

1.    Figure 1 and Figure 3 describe work using cerebral ECs isolated from Mgp+/+ and Mgp-/- mice, but the method for isolating these ECs and characterisation of these cells are not described in either the methods or the results. Please add protocols in the method section, and include in the result section the confocal images of isolated ECs immune-stained with CD31 (or equivalent EC marker) and MGP. 

2.    Results in Figure 2 does not provide sufficient data to prove that BMP6 is the cause for the observed Notch activation in cerebral ECs shown in Figure 1.

a.    In Figure 2a, the authors tested a panel of BMPs and concluded that only BMP6 induces Notch1, Jagged 1 and Jagged 2 expression. However, in their previous publication (ref 4, Figure 3A), they have shown that BMP9 can induce Notch1, Jagged 1 and Jagged 2 in the same cells using western blotting. Please explain why the results are different.

b.    In this figure, a 48-hour treatment was used for the signalling assay. Such condition is not suitable for addressing which BMP is the specific ligand and dissecting the receptor utilisation. This is because many secondary gene targets can be induced during such time frame which can then induce the observed signalling. To prove BMP6 is the only ligand for the observed effects, the authors should show that BMP6 can initiate signalling in these cells by Smad1/5 phosphorylation at 30-60 min time point, whereas other BMP ligands cannot. Also, the primary gene targets and secondary gene gargets should be monitored at 1 hour and 5 to 8-hour time point. 

c.    Fig 2, b-d, siRNA experiment, the knockdown efficiency should be included to support the expression levels of these receptors and knockdown efficiency in the experiment. 

3.    To support that BMP6 is the cause for the change of BMP signalling in Mgp-/- mice, the authors should provide data to compare the expression of the panel of BMP in the brain from control and knockout mice, as those shown in reference 5, Fig 4A.

In conclusion, without these additional data, the link and involvement of BMP6/ALK3 signalling cannot be justified.

Author Response

1.“Figure 1 and Figure 3 describe work using cerebral ECs isolated from Mgp+/+ and Mgp-/- mice, but the method for isolating these ECs and characterisation of these cells are not described in either the methods or the results. Please add protocols in the method section, and include in the result section the confocal images of isolated ECs immune-stained with CD31 (or equivalent EC marker) and MGP.”

We added the protocol for EC isolation in the Method section. Also, we added the co-immunostaining images of the endothelial marker vWF with MGP in Figure 1.

2.“ Results in Figure 2 does not provide sufficient data to prove that BMP6 is the cause for the observed Notch activation in cerebral ECs shown in Figure 1.

a.    In Figure 2a, the authors tested a panel of BMPs and concluded that only BMP6 induces Notch1, Jagged 1 and Jagged 2 expression. However, in their previous publication (ref 4, Figure 3A), they have shown that BMP9 can induce Notch1, Jagged 1 and Jagged 2 in the same cells using western blotting. Please explain why the results are different.”

Thanks for this constructive comment. To identify which BMP ligand and receptor are responsible for the induction of the Notch components, we treated HBMECs with different doses of BMP ligands (0-300 ng/ml) as used in previous studies (5, 19, 26-30), including BMP2, 4, 6, 7, 9 and 10. After 48 hour of treatment, we examined the expression of Notch1, and Jagged1 and 2, and found that BMP6 strongly induced these Notch proteins (Figure 2a). In addition, BMP9 mildly induced same Notch components (Figure 2a). To further differentiate the responses, we treated HBMECs with BMP6 (100 ng/ml) and BMP9 (10 ng/ml), respectively. The results showed that BMP-9 induced all the ligands and receptors of the Notch pathways, similar to our previous study (4), whereas BMP-6 only induced the expression of Notch1, and Jagged1 and 2 (Figure 2a-b). Indeed, the BMP-6 induction pattern of Notch1, Jagged1 and 2 in HBMECs was similar to the induction pattern in the cerebral endothelium of Mgp-/- mice.

We have added the data and updated text in the manuscript.

b.“ In this figure, a 48-hour treatment was used for the signalling assay. Such condition is not suitable for addressing which BMP is the specific ligand and dissecting the receptor utilisation. This is because many secondary gene targets can be induced during such time frame which can then induce the observed signalling. To prove BMP6 is the only ligand for the observed effects, the authors should show that BMP6 can initiate signalling in these cells by Smad1/5 phosphorylation at 30-60 min time point, whereas other BMP ligands cannot. Also, the primary gene targets and secondary gene gargets should be monitored at 1 hour and 5 to 8-hour time point.”

We treated HBMECs with BMP-6 and examined the phosphorylation (p) of SMAD 1/5/8 after 30 and 60 minutes of treatment.  The results showed that pSMAD1/5/8 was significantly increased at 30 minutes and even more at 60 minutes (Figure 2c), suggesting that early response to BMP-6 in HBMECs was mediated by pSMAD1/5/8. We then examined the Notch1 intracellular domain (NICD1) and found that NICD1 was significantly increased after 5 and 10 hours of BMP-6 treatment in the HBMECs (Figure 2d). The results suggested that BMP-6 induces Notch components through pSMAD1/5/8.

Even different BMPs activate different receptors, the same SMAD1/5/8 would be activated with different BMP treatment. Thus, we limit the treatment by only using BMP6 to avoid the complicity outcomes of phosphorylation of SMAD1/5/8 by different BMPs and receptors.

c.“Fig 2, b-d, siRNA experiment, the knockdown efficiency should be included to support the expression levels of these receptors and knockdown efficiency in the experiment.”

We have added the data to the manuscript.

3. “To support that BMP6 is the cause for the change of BMP signalling in Mgp-/- mice, the authors should provide data to compare the expression of the panel of BMP in the brain from control and knockout mice, as those shown in reference 5, Fig 4A.”

To assess the BMP levels in the Mgp-/- brains, we examined the expression of BMPs using real-time PCR and compared to wild type brain. The results showed no significant difference between wild type and Mgp-/- brains (Figure 2e). The results were consistent with our previous studies (5, 27, 29), which showed that lack of MGP increased the BMP activity, but not necessarily the expression. 

Reviewer 2 Report

Title: Crosstalk between BMP and Notch Induces Sox2 in 2 Cerebral Endothelial Cells

Summary:

In the present study Wu and Yao et al. investigated the signalling crosstalk between BMP and Notch pathway to induce Sox2 expression in brain endothelial cells (HBMECs). They showed that in cerebral endothelial cells isolated from Matrix Gla Protein (Mgp) wildtype and knockout transgenic mice Notch1, Jagged1 + 2 as well as Sox2 expression is enhanced. Jagged 1 + 2 increased Sox2 expression in cerebral endothelial cells, Jagged 1 + 2 knockdown decrease Sox2 as well as EC-specific Notch1 deletion in Mgp-/- background rescued Sox2 expression back to wildtype baseline. Different BMP ligands and siRNA-mediated receptor knockdowns were tested in HBMECs identifying BMP6 and ALK3 to regulate Notch1 and Jagged 1 + 2 gene expression. Finally Chip assays were performed with Notch signalling target transcription factor RBPJκ in cerebral ECs derived from Mgp+/+ versus Mgp-/-.

General comments:

The findings in the current version of the manuscript are not entirely novel, which reduces the overall interest to the readers. The authors have already shown in earlier publications that in brains from Mgp-/- mice and in HBMECs treated with siRNA against Mgp more Jagged 1 + 2 is expressed and more active Notch signalling is present {Yao, 2013 #3}. Here the authors have now isolated cerebral endothelial cells from Mgp-/- mice and report the same result.

Some important controls are missing - knockdown efficacy for siRNA experiments as wells as activation of signalling pathway.

Furthermore, conclusions drawn from the presented results are unprecise and therefore overinterpreted. For example page 5, line 137-139: “Together, we have identified that BMP6 induced Notch components to alter the complex of Notch-associated proteins in the Sox2 promoter and activate its expression in cerebral ECs. Only in Figure 2 in HBMECs a connection between BMP6, Alk3, Notch1 and Jagged1 + 2 is shown. However, only one concentration per BMP ligand was used and it is well-known that BMP ligands form gradients and act concentration-dependent. In Figure 3 and 4 the Mgp-/- mouse is used, but, Mgp inhibits several BMP ligands and does not mimic a BMP6 gain-of-function transgenic mouse. Finally, results obtained from the Chip assay should be discussed in light of several publications that have shown that BMP and Notch signalling synergistically interact to regulate the expression of target genes. {Itoh, 2004 #6}{Takizawa, 2003 #7}{Dahlqvist, 2003 #9}. The authors should carefully discuss and weight their conclusions as not only expressional regulation of Notch1 and Jagged 1 + 2 may play a role in transcriptional regulation of Sox2 but also direct BMP and Notch signalling interaction through Smad and RBPJκ.

Major comments:

Introduction:

1)      More precisely, page 2, line 47-48, ”Recent studies from our lab show… or cite someone else who has also shown this interaction.

Figure 2:

1)      Why are different concentrations of BMP ligands used? It would be better to use molarity to compare directly the number of the different BMP molecules. To draw the conclusion that BMP6 is the only BMP that induce Sox2 expression, concentration series for the other BMPs have to be performed to exclude that they also can induce So2 expression.

2)      Please show knockdown of siRNAs Alk2-7.

Figure 3:

1)      Please show some controls in IB for activation of Notch signalling pathway (3b) and knockdown for Notch 1-4 (3d). Preferably on the same membrane.

Methods:

1)      Methods description for isolation and culture of cerebral ECs are missing.

Minor comments:

Page 1, line 38, there is a typo in “activing receptor-like kinase”

Author Response

General comments:

“The findings in the current version of the manuscript ……   The authors should carefully discuss and weight their conclusions as not only expressional regulation of Notch1 and Jagged 1 + 2 may play a role in transcriptional regulation of Sox2 but also direct BMP and Notch signalling interaction through Smad and RBPJκ.”

We cited the literatures and added “The interplay between BMP and Notch signaling may alter the downstream effects of BMP activity (35-37), or change the Notch activity on target genes (35-37), at least in part through interaction between NICDs and SMADs (36, 37).” to discussion of the manuscript. 

We also added “We show that Sox2 is a direct target of Notch signaling in brain ECs and that the DNA binding of RBPJk in the Sox2 promoter of is elevated with excess BMP activity. Since pSMAD1/5/8 are also increased in Mgp-/- null ECs, the possibility remains that RBPJk interacts with the SMADs to regulate the Sox2 expression.” to the discussion of the manuscript.

Major comments:

Introduction:

1) “More precisely, page 2, line 47-48, ”Recent studies from our lab show… or cite someone else who has also shown this interaction”.

It has been updated.

Figure 2:

1)“Why are different concentrations of BMP ligands used? It would be better to use molarity to compare directly the number of the different BMP molecules. To draw the conclusion that BMP6 is the only BMP that induce Sox2 expression, concentration series for the other BMPs have to be performed to exclude that they also can induce So2 expression.”

To identify which BMP ligand and receptor are responsible for the induction of the Notch components, we treated HBMECs with different doses of BMP ligands (0-300 ng/ml) as used in previous studies (5, 19, 26-30), including BMP2, 4, 6, 7, 9 and 10. After 48 hour of treatment, we examined the expression of Notch1, and Jagged1 and 2, and found that BMP6 strongly induced these Notch proteins (Figure 2a). In addition, BMP9 mildly induced same Notch components (Figure 2a). To further differentiate the responses, we treated HBMECs with BMP6 (100 ng/ml) and BMP9 (10 ng/ml), respectively. The results showed that BMP-9 induced all the ligands and receptors of the Notch pathways, similar to our previous study (4), whereas BMP-6 only induced the expression of Notch1, and Jagged1 and 2 (Figure 2a-b). Indeed, the BMP-6 induction pattern of Notch1, Jagged1 and 2 in HBMECs was similar to the induction pattern in the cerebral endothelium of Mgp-/- mice.

The results have been added to the manuscript.

2)“Please show knockdown of siRNAs Alk2-7.”

We have added the data to the manuscript.

Figure 3:

1) “Please show some controls in IB for activation of Notch signalling pathway (3b) and knockdown for Notch 1-4 (3d). Preferably on the same membrane.”

We have added NICD1 as a control for activation of Notch1 to Figure 4b, and added data of efficiency of selected siRNAs to Figure 4d. 

Methods:

1)“Methods description for isolation and culture of cerebral ECs are missing.”

The method for EC isolation has been added to the manuscript.

Minor comments:

“Page 1, line 38, there is a typo in “activing receptor-like kinase”

It has been updated.

Round 2

Reviewer 1 Report

The authors have performed additional experiments and the manuscript has improved significantly. I do not have further comments.

Reviewer 2 Report

The authors have improved the paper adequately in response to the reviewers' comments.